# The Role of Epigenetic and Biological Biomarkers in the Diagnosis of Periodontal Disease: A Systematic Review Approach

**DOI:** 10.3390/diagnostics12040919

**Published:** 2022-04-07

**Authors:** Erin Faulkner, Adelaide Mensah, Aoife M. Rodgers, Lyndsey R. McMullan, Aaron J. Courtenay

**Affiliations:** 1School of Pharmacy and Pharmaceutical Sciences, Ulster University, Coleraine BT52 1SA, UK; erincf98@gmail.com (E.F.); Mensah-a@ulster.ac.uk (A.M.); 2Department of Biology, The Kathleen Lonsdale Institute for Human Health Research, Maynooth University, W23VP22 Maynooth, Kildare, Ireland; Aoife.Rodgers@mu.ie; 3DJ Maguire and Associates, 83 Bridge Street, Portadown BT63 5AA, UK; lyndseymcmullan@hotmail.com

**Keywords:** periodontal disease, epigenetic, biomarkers, systematic, review, exRNA, gingival crevicular fluid

## Abstract

The aim of this systemic review was to collate and analyze existing data from published literature sources to identify the current understanding of the role of epigenetic and biological biomarkers in periodontal disease and diagnostics. A comprehensive searching strategy was undertaken in Embase, Medline, The Dentistry and Oral Sciences and CINAHL databases. Grey literature searching strategies were also employed. Articles published in the English language between 2017–2020 were included. A total of 1014 studies were returned of which 15 studies were included. All included articles were cross-sectional, case–control studies. Relevant data were extracted according to various demographic and methodological factors including cohort size, oral biofluid sampled, number of examiners, smoking status and reported outcomes. A measure of the biomarker levels and corresponding significance were documented where possible. This review identified that exRNA has the greatest diagnostic potential, with four biomarkers (SPRR1A, lnc-TET3-2:1, FAM25A, CRCT1) displaying sensitivity of >71% and specificity of 100% in the assessed samples (*p* < 0.001) for gingivitis. This work also identifies the need for a unified approach to future research to draw meaningful comparison. Further investigations are warranted to definitively validate exRNA data and for the development of an exRNA-specific point-of-care diagnostic test.

## 1. Introduction

Oral diseases affect up to 3.5 billion people globally and periodontal disease (PD) is considered the sixth most common disease worldwide [1]. According to the British Dental Association, approximately 90% of the global population will experience loss of connective tissue attachment at some stage in their life [2]. Untreated PD can lead to aesthetic disruptions, tooth mobility, dysfunctional or absent mastication functions and ultimately, tooth loss. Systemic complications of PD for example, cardiac complications, low birth weight and glycemic control issues in diabetics also impose a burden on the health system [3]. A study conducted by Bahekar et al. found that those with less than ten teeth had a 1.24 times increased risk of developing coronary heart disease, when compared to those with more than ten teeth [4]. A further study concluded that the odds ratio of mothers with periodontitis and pre-term births or low birth rates was 2.83, compared to those that were periodontally healthy [5]. In 2016, the NHS had an average annual expenditure of GBP 3.4 billion for the commission of primary and secondary care dental services in the UK [6]. Such costs impart a huge financial burden on the overall healthcare system. Early detection and eradication of disease progression is thus financially attractive and improves patient outcomes.

PD involves the impairment of the periodontium, which includes the gingiva, periodontal ligaments, alveolar bone and cementum. This impairment is associated with microbial dysbiosis of oral the microbiota, resulting in detachment of tooth from the periodontium. Gingivitis, the most common and mildest form of PD, is characterized by the accumulation of bacteria in the gingival pocket, and subsequent plaque build-up on the tooth. Gingivitis arises as a result of poor oral hygiene, which may be improved with improved hygiene practices. Progression of gingivitis can lead to periodontitis, a chronic oral disease resulting from deep penetration of bacteria into the gingival pocket and surrounding periodontium. Bacteria within the gingival sulcus thrive on tissue proteins and alter the pH of the cavity, leading to the precipitation of calcium from Gingival crevicular fluid (GCF) and saliva, thus contributing to calculus production [7]. Dysbiosis within these oral tissues can result in inflammation. Such inflammatory responses include elevated levels of C reactive protein, a biomarker of inflammation. Moreover, increased production of pro-inflammatory cytokines such as interleukin IL-1 and IL-18 from inflammatory cells at sites of inflammation has also been demonstrated [8]. In addition, salivary matrix metalloproteinases levels increase through prostaglandin E2 in PD [9,10].

Diagnosis only often occurs following significant disease progression. It requires clinical examination of the periodontium, and a Basic Periodontal Examination (BPE) score assigned [11]. Clinical examination involves visual inspections of the periodontium and surrounding tissues for evidence of inflammation, including a color and contour margins of the gingiva. A BPE is then conducted which consists of probing the gingival crevice using a PD-specific probe and the depth of the sulcus, clinical attachment loss, furcation involvement, tooth mobility and degree of bleeding upon probing are noted. The results of the examination are then collated, and diagnosis is determined based on a standardized risk score chart and updated classification from the World Workshop on the Classification of Periodontal and Peri-Implant Diseases and Conditions 2017 [12]. Challenges for the clinician exist with the current means of diagnosis. Difficulties in maintaining consistent probing pressure, as well as angulation between individual probing sites, and indeed individual patients, is an issue for those assessing the periodontium [13]. The presence and degree of inflammation at the tissue site also influences probe depth. Inflammation will cause the probe tip to penetrate further into the gingival crevice, which does not correlate to clinical attachment loss. The physical act of probing the potentially damaged periodontium imposes the risk of negatively impacting the integrity of the gingiva and exacerbating symptoms further [14]. The use of electronic probing in periodontal disease has been reported to be a good alternative to manual probing techniques [15]. Radiographic assessment of bone loss also has its limitations, only detecting loss once 30–50% of alveolar bone loss is experienced [16]. This current diagnostic evaluation not only possesses limited diagnostic sensitivity and specificity but is also invasive, arduous and results are open to the interpretation of the diagnostician [17]. Improved diagnostic techniques are subsequently required to detect PD in its infancy and in a non-invasive, holistic manner.

Oral fluids such as saliva and GCF have potential diagnostic capabilities containing both local and systemic indicators of disease state [18,19]. Saliva is a complex system of sulcular fluid components as well as microorganisms, periodontium and systemic inflammatory products [20]. The abundance, speed and ease for samples collection make this an excellent target for diagnostic media. A possible challenge of saliva as a diagnostic medium is potentially inadequate sample volumes due to dry mouth or certain medication [21]. GCF is both a physiological fluid and an inflammatory exudate. The composition and volume of GCF differs between healthy and diseased patients [7]. Current GCF collection procedures impose several challenges. The collection is arduous and technically challenging, requiring specialized equipment. There is also the risk of the GCF sample becoming contaminated with blood or plaque. The presence of either would suggest further damage to the vulnerable tissue [22].

A biomarker, by definition, is “a naturally occurring molecule, gene, or characteristic by which a particular pathological or physiological process, disease, etc. can be identified” [23]. Ideal biomarkers are highly specific, sensitive, rapid, inexpensive to detect and predictive [24]. Traditional biomarkers such as proteins, metabolites and matrix-metalloproteinases (MMPs) may aid in PD diagnosis. Epigenetic biomarkers (agents that control the function of gene expressions and mediators) play a significant number of key roles in PD progression. For example, chronic inflammation in the periodontium results in the dysfunction of DNA methylation in genes which code for pro-inflammatory (TNF-alpha, IL-6) and anti-inflammatory (IL-4, IL-10) cytokines. Detection of such biological and epigenetic biomarkers in oral biofluid has the potential to diagnose PD in the early stages of disease progression [16]. Due to the complex interplay of inflammatory and regenerative pathways during periodontal disease progression, it is unlikely that one biomarker or epigenetic marker will ever be solely diagnostic. Indeed, it is key that the relationship between biomarkers present and the control of expression of proteins will be needed to fully diagnose and treat periodontal diseases.

This review utilized a systematic approach to collate and analyze existing data from published literature sources to identify the current understanding of the role of epigenetic and biological biomarkers in PD and diagnostics. Specifically, PICO modelling of the research question is defined and comprehensive searching strategies in Embase, Medline, The Dentistry and Oral Sciences and CINAHL databases are described. MeSH headings and keywords for articles published between 2017–2020, in the English language were used and a total of 1014 studies were returned. A PRIMSA model was followed, and 15 references were identified for inclusion in this systematic review, all of which were cross-sectional, case–control studies. A previously validated quality and bias assessment tool was applied to ascertain the validity of the included studies and identify the known risks in reported studies.

## 2. Materials and Methods

### 2.1. PICO Modelling

The purpose of this systematic review was to investigate “The role of epigenetic and biological biomarkers in the diagnosis of Periodontal Disease.” A diagnostic test for PD which is minimally invasive and has a high degree of sensitivity is currently warranted. The interventions reviewed include biomarkers and diagnostic tests specific to periodontal disease. The comparison in this systematic review was periodontally healthy subjects and periodontally diseased subjects, as classified by each study. The outcome was the identification of the most specific epigenetic and biological biomarkers and suggestion of possible diagnostic assay (s) for point-of-care diagnosis.

### 2.2. Search Strategy and PRISMA Statement

A robust literary search of four relevant databases was conducted from 27 September 2020 until 14 October 2020 to comprehensively analyze the study hypothesis outlined previously. Embase, Medline, Dentistry and Oral Sciences and CINAHL databases were chosen as they published the most suitable articles for this research. The databases were all searched meticulously. Medical Subject Headings (MeSH) and subject terms were used in the applicable databases, as well as keyword searches which remained constant across the four databases. Manual searching of Google Scholar and the Cochrane Library was also conducted to identify any pertinent research not included in the databases. A grey literature search was undertaken in OpenGrey to identify unpublished literature relevant to this review. Reference lists of applicable articles were also reviewed to obtain further studies. Boolean operators, “AND” and “OR” were utilized to achieve precision in search results while the truncation, “*” was used to allow for variations in spelling. An English language limit was applied in conjunction with a date limit of 1 January 2017–14 October 2020. Search results from each database were recorded and can be seen in Appendix A. All search results were exported to ProQuest RefWorks 2.0 where duplicate articles were removed. The articles were then initially assessed by title, abstract and keywords for inclusion in this systematic review. Articles which could not be excluded based on title, abstract or keywords alone were reviewed in full before a final decision was made by the author. Reasons for exclusion were outlined in Table 1 listed below. After the initial assessment, the full text of remaining articles was appraised in detail to assemble the final collection of papers for inclusion in this systematic review. The Preferred Reporting Items for Systematic Reviews and Meta Analysis (PRISMA) 2020 guidelines were used; however, the study was not registered.

### 2.3. Study Selection 

Eighty-nine articles were reviewed for full-text assessment. Articles were read in their entirety and assessed based on the inclusion and exclusion criteria as outlined in Table 1. Those which were deemed unsuitable were removed from the article list and reasons for exclusion were noted. A final collection of 15 articles were printed to carry out an in-depth assessment of contents and prepare for data extraction. 

As previously stated, an English language limit was applied to studies as translation was beyond the remit of this systematic review. Studies published within the last three years were analyzed as the World Workshop on the Classification of Periodontal and Peri-Implant Diseases and Conditions was held in 2017, which reclassified periodontitis in conjunction with the American Academy of Periodontology and the European Federation of Periodontology, an update from the 1999 classification. Therefore, a time limit of 2017–2020 was decided in order to analyze the most up to date literature on the topic at hand. Studies focusing solely on human participants were included and animal studies were excluded due to the differences in human and animal physiology. Peri-implant studies were excluded, as only studies comparing periodontal health and disease in those with natural teeth were analyzed. In order to conduct a robust, inclusive systematic review of the current literature, references analyzing both male and female participants were included, while gender-specific studies were excluded. Studies examining exclusive sub-populations of participants for example by ethnicity, smoking status, patients with diabetes or patients who were considered immunocompromised were excluded, as such studies often analyze genomic specificities in relation to disease state. The exclusion of such cohorts resulted in the extensive examination of disease state in systematically healthy patients to produce a review which dealt with biomarker analysis holistically. While the etiology of periodontitis remains constant between adults and children, there are various factors contributing to the disease development in children. Hormonal fluctuations during puberty can contribute to the disease progression [25]. Such fluctuations can also contribute to gingivitis development in pregnancy and lactation, as estrogen and progesterone levels are increased in gingival tissues [26]. Current PD diagnosis also differs between adults and children [13]. Due to these additional contributing factors, participants under the age of eighteen, pregnant and lactating women were excluded from this systematic review. Drug-induced, experimentally induced disease and tissue culture analysis references were excluded as this systematic review aimed to observe disease diagnosis in naturally occurring disease. Systematic review, literature reviews, abstracts and editorials were excluded as often these forms of analysis provide a limited assessment of the literature and addition of previous systematic reviews in a systematic review can artificially inflate the knowledge base [27].

### 2.4. Data Extraction 

Following article selection, data extraction was undertaken. The following data were extracted from the reference collection: Author;Year of publication;Location of publication;Study design;Number of participants;Fraction of participants both male and female;Mean age and standard deviation of participants;Classification of disease;Biomarker analyzed;Biofluid analyzed;Biomarker analysis method;Method of disease classification;Number of assessors of disease state;Method of sample collection;Outcome of studies.

Quality assessment and risk of bias of each study was conducted following the Cochrane Handbook for Systematic Reviews of Diagnostic Test Accuracy, which follows a traffic light ranking system [28]. This assessment tool was chosen as all fifteen articles included in this study were cross-sectional, observational studies. Assessment of the chosen articles was based on the population spectrum, study design, study conduct, reporting of results and sponsoring. Each study was assessed based on each of the eleven quality parameters and assigned a red, yellow or green light for ease of visual assessment. Red lights were allocated to those which did not comply with the outlined bias standard, yellow lights for those which were unclear and a green light for those which complied with the bias standard. The quality assessment and risk of bias assessment was undertaken by the reviewer. 

### 2.5. Statistical Analysis

An analysis of the statistical methods adopted throughout the reference collection was conducted to determine the appropriateness of methods used and to evaluate the results found. Comment was made on the methods used and the outcomes of this analysis are detailed in the *Result* of this systematic review.

## 3. Results

The literature search yielded a total of 1014 results. Following deduplication, a total of 928 references were identified for assessment. These articles were then screened by both title and abstract to determine their suitability for this systematic review. Following this assessment, 839 articles were deemed unsuitable for inclusion based on the previously outlined inclusion and exclusion criteria as listed above. Those which could not be excluded based on title or abstracts screening alone were flagged for full text review, of which, 89 references were identified. Following full text review, 15 articles were identified as suitable for inclusion in this systematic review. Figure 1 illustrates how the final collection of articles were curated. 

All 15 articles included in this systematic review were identified as case–control cross-sectional observational studies. 

### 3.1. Quality Assessment and Risk of Bias

The quality assessment and risk of bias described previously are outlined in Figure 2. Overall, the results of the quality assessment and risk of bias yielded positive results; however, there were a few points worth noting. Of the 15 cross-sectional studies, all 15 avoided differential verification and incorporation bias. This review found that three of the studies did not comprehensively represent the target population. Sai Karthikeyan et al. included participants between the ages of 21 and 65 years old in the study. Micó-Martínez et al. sampled GCF in a population ranging from 25 years old to 61 years old. Cherian et al. included those from 18–45 years old. Each of these studies omitted populations outside these age ranges which could benefit from epigenetic biomarker evaluation for the diagnosis of periodontal disease. From the analysis it was concluded that five studies did not have an appropriate reference standard, while one article included had an unclear reference standard. The five studies that did not have appropriate reference standards used multiple examiners to carry out the classification examinations of participants therefore, discrepancies in results may have occurred. The one article whose appropriateness was unclear did not disclose how many examiners undertook the classification examination. There were 11 articles which did not disclose the time frame between classification of disease and sample collection. It was identified that four articles noted the time delay and rationale for such a limit. In the study carried out by Inönü et al., 10 participants were unaccounted for. As such, the study exhibited partial verification bias and did not correctly explain withdrawals from the study. Sample collectors and analysts were blind to the reference standard results in three of the studies which limited any influence on the readings of biofluid analysis. While 13 articles blinded the index test results. The reference standards blinding process was not disclosed in twelve of the studies, while blinding of index test results was not disclosed in two studies. Accordingly, they may have been subject to diagnostic review bias. Uninterpretable results were not explained in eight of the articles. Contaminated or inadequate samples of oral biofluid were not referred to. Sponsoring was not disclosed in three of the references leading to a lack of transparency in these studies. 

### 3.2. Summary of Tables

The reference list underwent a thematic analysis and was divided into subgroups based on the type of biomarker analyzed, as decided by the author. There were four subgroups established: RNA biomarkers;Protein biomarkers;Metabolite biomarkers;Inflammatory biomarkers.

The basic study demographic characteristics, classification of disease, biomarker analyzed and biofluid sampled are listed in Table 2, Table 3, Table 4 and Table 5. 

Classification methods, sample collection, sample analysis and outcomes of the studies are also listed in Table 6, Table 7, Table 8 and Table 9.

In studies where the distribution of male and female participants between disease states was unavailable, the number of participants in each disease group was given or the overall number of male and female participants. In studies which provided median age or age range, these figures were documented and noted. References which were composed of two phases were also noted. Classification of disease was recorded as a standard reference guide. Mean values of classification parameters were recorded among disease states for studies which did not provide classification criteria. These studies were highlighted. In studies which also analyzed other physiological fluid samples such as serum or blood for diagnostic biomarkers, only data for oral biofluid were extracted.

### 3.3. Age

Participant age was recorded as mean values and standard deviation in 13 of the studies. An age range was recorded in one study [29], while another recorded the age of participants as a median figure [30]. Ages were matched between healthy and diseased participants, within 5 years, in six of the studies, while the participants classified according to a disease state were older than the control group in eight of the studies. As Cherian et al. only provided an overall age range of participants, it is unclear as to whether participant groups were age matched. 

### 3.4. Classification

The fifteen studies analyzed studied the role of biomarkers in six different forms of periodontal disease. The classification of disease state and number of studies analyzing each state is outlined in Figure 3. Probing depth (PPD) and clinical attachment level (CAL) were measured in all sixteen studies. Bleeding on Probing (BOP) was a means of diagnosis in 13 of the studies. Plaque Index (PI), Gingival Index (GI), Body Mass Index (BMI) and Supragingival Calculus (SC) were parameters also employed throughout the studies. It was noted that Chatzopoulos et al. analyzed participants with both a “normal” classification and “strict” classification. The criteria for both classification models were outlined in Table 7. The analysis revealed that nine studies disclosed the classification system which was followed to categorize participants; eight studies followed The Classification of Periodontal Diseases and Conditions Armitage 1999, while one followed the World Workshop on the Classification of Periodontal and Peri-Implant Diseases and Conditions 2017. The remaining six studies did not stipulate which guide was followed. Of the 15 articles reviewed, 8 employed 1 single examiner to measure the clinical parameters, 4 studies employed more than 1 examiner, and the remaining 3 references did not specify how many examiners undertook the clinical examinations.

### 3.5. Biofluid Samples

This systematic review focused on the role of biomarkers in oral biofluids, saliva and GCF. Of the fifteen studies included in this review, seven analyzed saliva samples, six analysed GCF samples and two studies collected both saliva and GCF samples. Both stimulated and unstimulated saliva samples were collected throughout the nine studies. Oral hygiene practices, eating and drinking were asked to be avoided for a period of one hour in three studies, two hours in four studies and for a period of 24 h in one study. Only one study did not specify such a requirement. A total of six studies collected the saliva samples before midday. Collection in the morning period was rationalized by Bostanci et al. who stated that in the morning, the composition of saliva was least variable. 

GCF samples were collected by three different methods. Of the fifteen studies, five of those used PerioPaper^®^ strips, one study used paper points and one studied used micropipettes. There was no evidence suggested in the studies as to why any specific method was preferred. Contamination of samples was alluded to in six of the studies. Cotton rolls were used to limit saliva contamination, while those contaminated with saliva or blood were discarded and the samples were re-collected.

Samples of biofluids were pooled in two studies. Hartenbach et al. pooled saliva samples from pairs of participants in each group based on age. In the study carried out by Nalmpantic et al., four GCF samples were acquired from each individual and these were pooled together. Pooled samples reduce cost and often lend to increased efficacy in analysis when compared to single samples [31].

### 3.6. Order of Methods

The order in which disease classification was conducted and samples were obtained varied throughout the studies. Sampling was carried out prior to clinical examination in six of the studies, while the opposite was true in three studies. The order of events was not described in the remaining six articles. Sai Karthikeyan et al. and Romano et al. both stated that sampling was conducted 24 h after examination to prevent false readings of samples and limit blood contamination. None of the six studies that obtained samples prior to clinical examination provided a rationale for this order of events. 

### 3.7. Biomarkers

Four categories of biomarkers were analyzed in this systematic review. 

#### 3.7.1. RNA

Both Mico-Martinez et al. and Kaczor-Urbanowicz et al. produced studies reporting significant findings based on the use of RNA in PD diagnosis. Mico-Martinez et al. concluded that miR-1226-5p had a 15.8-fold downregulation in those with CP compared to healthy participants. Kaczor-Urbanowicz et al. found four exRNAs (SPRR11, Inc-TET3-2:1, FAM25A and CRCT1) had the potential to provide significant discriminatory effects with sensitivity of 71% and 100% specificity in subjects with G. 

#### 3.7.2. Proteins

Throughout the four studies analyzing protein biomarkers in disease detection, eleven proteins were identified as being highly significant. 

The two-phase study carried out by Bostanci et al. identified five proteins with significant distinction between health and disease. MMP9, RAP1A and ARPC5 were all upregulated in disease while CLUS and DBMT1 were downregulated in those with disease. The authors deduced that the combination of ARPC5 and CLUS when tested together produced the greatest predictive power of disease.Chatzopoulos et al. examined the levels of SOST, WNT-5A and TNF-α. All three biomarkers were increased in disease compared to health. The “strict” periodontitis subgroup produced median levels of SOST (140.00 pg), WNT-5a (2.4 pg) and TNF-α (44.7 pg) compared with the “strict” healthy group who exhibited levels of SOST (78.9 pg), WNT-5a (1.29 pg) and TNF-α (27.3 pg). SOST was the only biomarker to produce results which demonstrated significant difference between both groups (*p* = 0.002). Both WNT-5a and TNF-α had *p* = 0.075 and *p* = 0.226, respectively.Malondialdehyde (MDA) levels in H, G and P were examined in the study by Cherian et al. Mean levels of MDA were significantly different between H (89.45 µM/100 mL) compared to P (281.58 µM/100 mL) with a *p* < 0.001.Hartenbach et al. concluded levels of Histatin-1, salivary acidic proline-rich phosphoprotein and cystatin-SA were increased in those with periodontitis.

#### 3.7.3. Metabolites

Bulleted lists look similar to this:Chen et al. identified 20 metabolites with *p* < 0.05 and Variability Importance Projection (VIP) >1. However, eight metabolites were established as having the greatest predictive power in distinguishing periodontal health from disease. Appendix A illustrates the metabolites identified along with their corresponding Fold Change and VIP.Similar to Chen et al., Pei et al. found 17 metabolites associated with periodontal health and disease with *p* < 0.05 and VIP > 1. Appendix A below depicts the most periodontally significant metabolites identified. Analysis of combinations of these metabolites revealed the pair of metabolites with the greatest predictive power for periodontal disease was n-carbamylglutamate and citramalic acid.The study by Romano et al. found nine significant metabolites to have discriminative capabilities between periodontal health and disease, analysis of which are outlined in Appendix A.Rzeznik et al. conducted a study which identified 11 metabolites with significant discrimination between periodontal health and disease. Five metabolites with *p* < 0.2 were analyzed and three metabolites, GABA, 1-Butyrate and Lactate, were concluded to be the main diagnostic biomarkers in the study. Appendix A outline the statistical relevance of these metabolites.

#### 3.7.4. Inflammatory

Bulleted lists look similar to this:

There were 13 inflammatory biomarkers identified across the 5 studies included in this systematic review.

Hong et al. analyzed the relevance of eight different inflammatory biomarkers in both GCF and saliva. The study concluded that MMP-8, MMP-9, MPO and Cystatin C were the most significant biomarkers for the discrimination of periodontal health from disease with sensitivity of 87.0%, 73.9%, 87.0% and 72.5%, respectively. MMP-8 and MPO displayed the greatest sensitivity towards gingivitis.Del-1, IL-17 and LFA-1 levels in G, CP and GAP were analyzed by Inönü et al. Del-1 levels were seen to be increased in both H and G compared to CP and GAP, while the opposite was true for IL-17 and LFA-1 levels. When all three biomarkers were analyzed for discriminatory significance, and ROC value of 0.893 was produced along with sensitivity and specificity both reading 83.3%.It was determined by Nalmpantis et al. that azurocidin levels in pooled GCF samples were significantly increased in those with CP when compared to the participants with healthy tissue, *p* < 0.001.Tasdemir et al. reported the median levels of suPAR, Galectin-1 and TNF-α in both GCF and saliva. suPAR levels were significantly increased in both G and P compared to H in both GCF and saliva samples.CD163 levels in GCF were measured by Sai Karthikeyan et al. The mean levels of CD163 in H, G and *p* were 30.49 pg/mL, 38.93 pg/mL and 59.81 pg/mL, respectively. The study concluded that increased levels of CD163 positively correlate to progressive disease state.

### 3.8. Smoking Status

Although articles analyzing smokers as individual participant groups were excluded from this systematic review, articles which included smokers as part of the participant groups were analyzed. Smoking status, current or past was referred to in three of the articles while smoking was an exclusion criterion in seven of the articles. Smoking status was not alluded to in five of the articles. 

## 4. Discussion

This review synthesized data from 15 studies all of which analyzed the role of biomarkers, found in saliva or GCF or both, to aid in the diagnosis of PD. Across the 15 studies, the levels of 51 biomarkers were measured in 1178 participants. This was the first systematic review to analyze the role of both biological and epigenetic biomarkers in oral biofluids and correlate the relevant data. The outcome of this systematic review revealed that biomarker analysis is a hugely promising field in the diagnosis of PD; however, improved study designs, with larger cohort groups, are required in order to develop a PoC diagnostic tool to aid in clinical decision making and management. 

### 4.1. Strengths and Weaknesses of Studies

#### 4.1.1. Study Design

All fifteen studies included in this systematic review were cross-sectional or case–control studies. Cross-sectional studies measure the prevalence of outcomes over a defined time period. Case–control studies analyze a disease state and participant past exposure which is then compared to a control group [32]. Table 2, Table 3, Table 4 and Table 5 above outlined the spread of study design among the final collection of articles. Participant selection bias was evident in some cross-sectional studies as seen in the study by Rzeznik et al. where the control group was composed of staff and students working within the study facility. Such bias negatively impacts on the quality of the study due to the distortion of results. Inönü et al. did not disclose how participants were selected thereby reducing the integrity of the study. Bostanci et al. conducted a two-phase cross-sectional case–control study whereby proteomics found in salvia were discovered in the initial phase followed by the validity of such proteins and their discriminatory capabilities in the second phase. The robust nature of this study imparts confidence in the validity of the results. A similarly robust study was conducted by Kazcor-Urbanowicz et al. who adopted a PRoBE design. This study type produces results which are evaluated by analysts who were blind to the disease classification phase. Subjectivity in results analysis is therefore eliminated increasing the acceptability of study findings among the dental and diagnostic fields [33]. Improved study design in future research would be advantageous for further development in this field.

#### 4.1.2. Study Size

The size of a study influences the reproducibility of results on a local, national or international scale. Small sample sizes can lead to the generation of false positives or negatives due to the limited portion of a population sampled. Another issue may be an inadequate population representation, rendering the research redundant. A population sample that is too large will lead to ethical implications as members of the public may be subjected to unnecessary medical intervention [34]. Inequality between sample sizes also impacts studies due to the under-representation or over-representation of particular groups. Micó-Martínez et al. sampled a population of 18 participants, resulting in under-representation of the target population thereby reducing the validity of the findings. Inequalities between diseased and healthy groups were evident in four studies [30,35,36,37]. As a result of under-representation of diverse populations, drawing conclusive and translatable conclusions from these studies proved difficult. Therefore, additional studies with inclusive population groups may contribute to improved quality and generalizability of this research.

#### 4.1.3. Age 

As this systematic review sought to comprehensively analyze the hypothesis in adults, studies analyzing those over the age of 18 were included in this review. There were three studies identified which did not comprehensively analyze all those over the age of 18. Cherian et al. analyzed those in the age range of 18–45; therefore, they did not consider those over 45 suffering from PD. Micó-Martínez et al. included participants between the ages of 36–61 only, while Sai Karthikeyan et al. analyzed participants between the ages of 21–65. Omission of participants outside of the defined age ranges led to representation bias. Age matching between diseased and healthy participants is often carried out in cross-sectional and case–control studies and was evident in six studies [36,37,38,39,40]. This process can improve the efficacy of the studies; however, the lack of age matching in studies does not diminish the overall efficacy of a study. 

#### 4.1.4. Classification

Discrepancies between the classification of disease state posed the greatest challenge for systematic revision when comparing the individual studies. Table 2, Table 3, Table 4 and Table 5 outlined the number of examiners employed to classify disease state in each of the studies. Those which employed a single calibrated examiner to classify the disease state of participants achieved accuracy in results due to the technique remaining constant thereby, contributing to an enhanced quality of work. The studies which employed more than one independent examiner possibly encountered inter-examiner discrepancies when carrying the examination technique which may have led to clinically altered classification [38,39,40,41,42]. However, an ideal classification process would adopt a fully blinded clinical cohort and provide secondary examiners in order to confirm the classification results achieved. Those which did not disclose the number of examiners lacked transparency thereby minimizing the reproducibility of their findings [29,30,43]. Chatzopoulos et al. classified participants according to two classification models; the “normal” classification and “strict” classification which was made up of a subgroup of twenty participants from the “normal” group. This division of classification allowed for internal study comparisons contributing to improved reproducibility of results. The classification systems used also differed between the studies. There were eight studies which followed The Classification of Periodontal Diseases and Conditions Armitage 1999. One study followed the updated 2017 system [37]. The remaining studies did not disclose which system was used, which resulted in further issues when comparing studies. Uniform classification systems between studies would improve the overall efficacy of results.

#### 4.1.5. Order of Methods

As the current diagnosis of PD involves physically examining potentially inflamed tissue, the order of classification and sample collection within the studies was hugely important when analyzing the results. The six studies which conducted the biofluid sampling prior to disease classification produced the most accurate readings as irritation of the tissue did not occur [29,30,35,36,38,44]. When the sample collection proceeded disease classification, the potential for localized inflammation and subsequent inflammatory biomarker production was increased [42,43,45]. Future studies would be improved by collecting samples prior to tissue examination. 

### 4.2. Significant Evidence

#### 4.2.1. RNA

Epigenetic diagnostic profiling displays huge potential in the diagnosis and monitoring of many diseases including PD. RNA can provide direct information regarding mediator activity pertinent to specific disease states and progression [46]. This ability to provide precise quantitative analysis has the potential to drastically improve patient outcomes and clinical decision making when it comes to periodontal health [47]. RNA biomarkers analyzed in two studies displayed statistically significant diagnostic potential. Kaczor-Urbanowicz et al. found four exRNA biomarkers (SPRR1A, lnc-TET3-2:1, FAM25A, CRCT1) with 71% sensitivity and 100% specificity (*p*-value < 0.001) for discriminating gingivitis from periodontal health. This study also discovered five mRNAs with diagnostic potential in various other fields of medicine. The study was conducted using a robust design and the results produced were highly reproducible, displaying the greatest diagnostic performance of the studies included within this systematic review.

miRNA-1226 was identified by Micó-Martínez et al. as significantly relevant in CP diagnosis (*p* = 0.0004). miRNAs are very attractive diagnostic biomarkers due to their high stability in biofluids commonly assayed using analytic techniques [43]. miRNA-1226 was downregulated in CP compared to periodontally healthy participants. This biomarker’s function is to target proteins, Calreticulin and Mucin-1, which have roles in calcium storage, gene expression regulation, genotoxic stress regulation and tumor suppression. Therefore, this downregulation has negative impacts on tissue composition on a molecular level, in particular, osteoblast and osteoclast function [48].

#### 4.2.2. Proteins

Proteomic research of saliva has been a promising field for PD diagnosis due to the systematic, whole mouth analysis of salivary biomarkers. Many sophisticated analytic techniques have been employed to detect these biomarkers such as Liquid Chromatography Mass Spectrometry, QTRAP Mass Spectrometry and High-Performance Liquid Chromatography which return highly accurate results [35,44]. Upon proteomic analysis of whole saliva, Bostanci et al. concluded that five biomarkers (MMP9, RAP1A, ARPC5 CLUS and DBMT1) held the greatest degree of accuracy in differentiating disease and healthy tissue and identified ARPC5 and CLUS as the protein pair with the highest degree of predictability. Paired biomarkers analysis improves the accuracy of results [49]. Sclerostin (SOST) levels in GCF were shown to significantly discriminate CP from periodontal health (*p* = 0.002) [36]. Cherian et al. found that MDA levels in saliva were significantly increased in participants with P compared to the control group, (*p* < 0.005); however, MDA levels could not significantly discriminate G from periodontal health. The proteomic analysis conducted by Hartenbach et al. uncovered 473 proteins in the saliva samples of both CP and healthy subjects. However, they did not uncover significant evidence to suggest that any specific protein can effectively discriminate CP from healthy tissues. 

#### 4.2.3. Metabolites

Metabolomic research, similar to proteomics, is an ever advancing and attractive area of diagnostics. Metabolomics holds many advantages such as large throughput capacities, precision and high sensitivity [50]. It was found that the pyrimidine metabolism pathway was the most significant in GCP and a combination of citramalic acid and *N*-carbamyglutamate provided adequate precision in GCP diagnosis [51]. It was concluded by Chen et al. that Glycine-d5 levels were decreased in those with GAgP; however, Pei et al. found increased levels of the metabolite in GCP. Inconsistencies between the studies may have been a result of variations between classification methods; Chen et al. did not state how many examiners conducted the clinical examination. The findings by Pei et al. were thought to be more substantial due the larger sample size and more significant results with VIP of 2.688 and *p* < 0.0001, (Chen et al. VIP = 1.14, *p* = 0.001). Within all four of the studies, limitations and insufficient evidence to significantly discriminate health from disease revealed further analysis of metabolomics within saliva and GCF are required in order to achieve clinical utility. 

#### 4.2.4. Inflammatory

The discriminatory effect of various inflammatory biomarkers in PD has been previously studied. Hong et al. claimed to publish one of the first studies identifying discriminatory biomarkers for G. It was found that MMP-8 and MPO produced the greatest discriminatory capabilities between G and healthy tissue. The utility and contribution of active MMP-8 (aMMP-8) in the classification of PD and on POC has been assiduously investigated and has impacted periodontitis and peri-implantitis disease classification [52,53,54,55]. Levels of aMMP-8 in oral rinses have been used to distinguish G [53] peri-implant disease [54] and severe periodontitis [52]. These studies confirmed the accuracy and specificity of aMMP-8 in reflecting the severity and progression of PD as compared to traditional diagnostic techniques.

Inönü et al. found that salivary Del-1 levels produced significant results differentiating healthy tissue and disease. However, similar to Hong et al., could not conclude the validity of these results. CD163 was found to hold PD diagnostic capabilities, as levels significantly increased from healthy tissue to disease, tagging CD163 as a promising periodontitis biomarker [38]. Although Nalmpantis et al. could not conclude with significant confidence that azurocidin could be used as a biomarker for diagnosing CP, the study found that the ELISA analytical technique displayed a high degree of diagnostic capability. SuPAR and galectin-1 were both found to have potential diagnostic proficiency [40]. However, similar to the other four studies, concluded that the biomarkers analyzed may facilitate in the future evaluation of PD diagnostic markers. Inflammatory biomarkers with significant diagnostic performances were not found in any of the five studies outlined. CD163 was found to have promise; however, the validity of such results may be subject to criticism due the quality of the study. 

### 4.3. Context for Practice

PD presents in approximately 40% of adults in the UK [56]. The complexity of the complications of PD vastly outweighs the relative ease at which the disease can be treated, which include improved dental practices, dental scaling and in some cases antibiotics [57]. The current diagnostic techniques measure past periodontal deterioration and therefore lead to the possibility of a delayed or inaccurate diagnosis [58]. Biomarker analysis displays a high degree of specificity, sensitivity and precision for disease states. Such techniques can diagnose PD in its infancy therefore improving patient outcomes and decreasing the burden on the healthcare system [59]. Currently, a Point-of-Care (PoC) diagnostic test exists which tests for MMP-8 levels in mouth rinse. The PerioSafe^®^ test offers sensitivity of 76–83% and specificity of 96% for the detection of MMP-8 in saliva, providing promising potential to aid in the diagnosis of PD [60]. However, this test does not provide accurate results in those with systemic disease, receiving active orthodontic treatment or those suffering from ulceration of the mouth. Another drawback of this test is its use of only a single biomarker. The analysis of a combination of biomarkers provides for increased accuracy and improved diagnosis [61]. Therefore, a PoC with high sensitivity and specificity, analyzing a combination of biomarkers specific to PD is required to aid in the clinical diagnosis of this inflammatory disease.

### 4.4. Review Limitations

An English language limit was applied to this systematic review, leading to the exclusion of articles in languages other than English. This limit was applied as English was the only language in which the author was fluent; however, suitable and valuable research published and not translated to English may have been omitted. There was also a three-year time frame applied to the searches as the World Workshop for the Classification of Periodontal and Peri-Implant Diseases and Conditions updated the classification system in 2017; therefore, only studies within this period were included, excluding other relevant research which may have been conducted prior to 2017. However, after analysis of the reference list, it was found that seven studies followed the 1999 classification system while only one study followed the updated system. Variations between classification guides, imposed difficulties in the cross comparison of studies. Applying such limits may have resulted in the omission of relevant studies.

The outlined databases were robustly searched while grey literature was found through searching in the relevant resources to include the largest cohort of relevant studies possible. These databases were decided by the author alone who deemed these most suitable; however, it must be recognized that other relevant data may have been omitted. There were three MeSH terms and eight keywords applied to the database searches along with suitable Boolean operators and truncation, outlined in Appendix A, which were decided by the author. No studies conducted in the UK or Ireland were returned after reviewing the full-text articles; however, the majority of research was not defined to a specific geographical area.

### 4.5. Future Recommendations

This systematic review identified the need for a more unified approach within the studies published on this subject area. Improved study designs adopting suitable follow up, appropriate blinding among analysts, larger diversity among participants, appropriate classification and the adoption of the updated World Workshop on the Classification of Periodontal and Peri-Implant Diseases and Conditions 2017 system of PD would greatly improve the quality of future studies in this field. These improvements would thereby provide conclusive, translatable evidence for the suitability of the biomarkers in question to aid in PD diagnosis. PoC diagnostics is the future of PD diagnosis and holds many advantages such as limiting the invasive nature of current diagnostic examination and indeed the need to collect blood samples thereby reducing the time required to achieve diagnosis. However, challenges also exist with the development of PoC tests. Such tests must be corroborated along with clinical examination and therefore the acceptance of these PoC tools by clinicians and dentists is of great importance but may prove difficult [22]. Therefore, education programs for dentists and clinicians highlighting the benefits of these tests may be employed to reinforce the need for improved diagnostics. Currently, salivary PoC technology exists (Oral Fluid Nanosensor Test) which can identify four mRNA species (SAT, ODZ, IL-8 and IL-1b). This technology employs micropatterned electrons and electric-induced deposition to sensitively detect specific mRNA levels [62]. Such technology could be adapted to detect specific exRNA species analyzed in this systematic review in order to develop a robust, efficient PoC diagnostic aid. 

## 5. Conclusions

The role of RNA, protein, metabolite and inflammatory biomarkers was analyzed across fifteen studies. The aims of this systematic review were to identify biomarkers with the greatest degree of sensitivity and specificity and identify biomarkers which may be used in a PoC test for PD. It can be concluded that the aims and objectives were achieved; within the limitations of this systematic review, exRNAs displayed the greatest promise, with a high degree of sensitivity (71%) and specificity (100%) for the detection of PD in adults. The studies included in this review also found that metabolite biomarkers show the least potential for discriminating disease from healthy tissue compared to the other biomarkers studied. The need for a unified approach to study design within this field of research was identified along with recommendations for future research. Significant research and development have been conducted to date; however, further analysis is required to develop an appropriate PoC assay for the detection of exRNA in oral biofluids. Epigenetic and biomarker diagnostics have advanced significantly, almost to the point of clinical usefulness. However, biomarkers of disease have largely been assessed in isolation. Moving forward researchers need to work collaboratively with clinicians, patients and industrialists to realize the diagnostic potential of epigenetic and biomarker profiling—considering a range of biomarkers in tandem, and no longer working in isolation. 

## Figures and Tables

**Figure 1 diagnostics-12-00919-f001:**
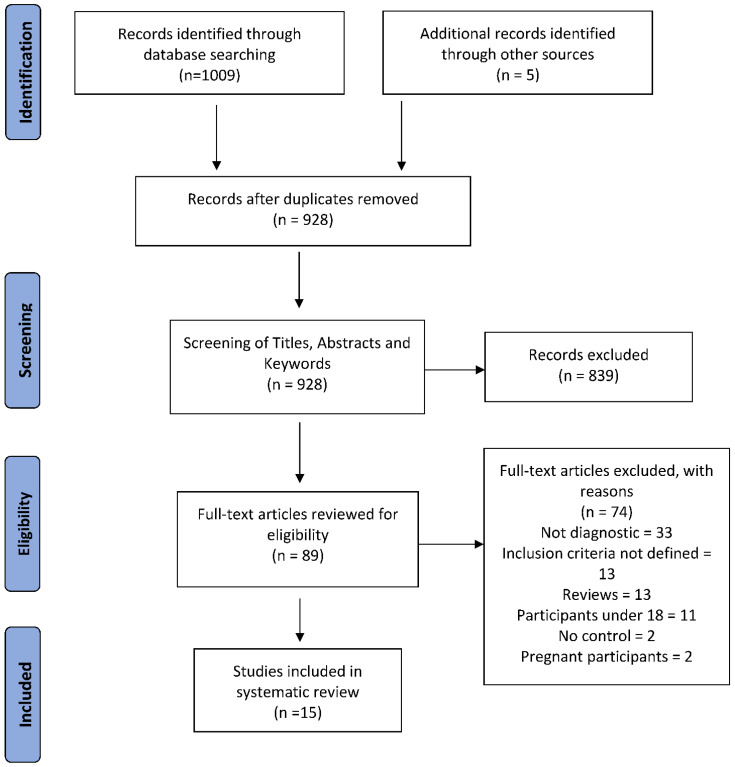
PRISMA diagram illustrating the method of article review and selection for inclusion in this systematic review. *n*, Number of articles included.

**Figure 2 diagnostics-12-00919-f002:**
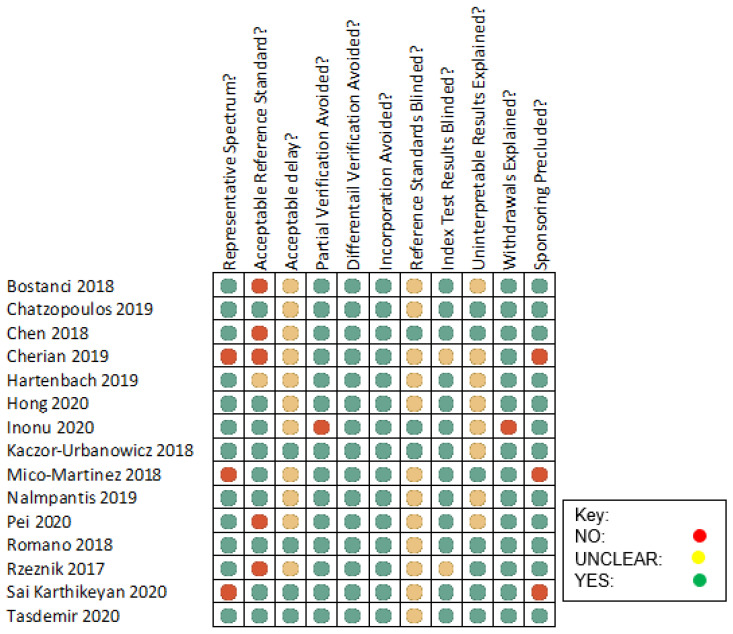
Assessment of quality and risk of bias of cross-sectional studies included in this systematic review.

**Figure 3 diagnostics-12-00919-f003:**
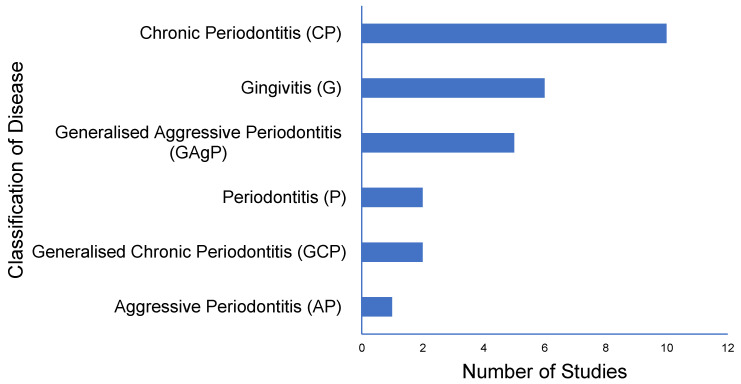
Horizontal bar chart outlining the breakdown of disease classification among the 15 studies.

**Table 1 diagnostics-12-00919-t001:** Inclusion and exclusion criteria for articles reviewed from database searching.

Inclusion Criteria	Exclusion Criteria
English language Published from 2017–2020 Human participants Male and female participants >18 years old Periodontal disease, gingivitis, periodontitis disease states Gingival crevicular fluid or salvia analysis	Participants under 18 years old Animal studies Peri-Implant disease Gender specific studies Studies looking at populations in isolation No control groups Studies comparing smokers versus non-smokers as independent study groups Studies focusing on groups with systemic conditions, i.e., Diabetes Miletus Pregnant and lactating participants Tissue culture analysis Drug-induced or experimentally induced disease Systematic reviews, literature reviews, abstracts and editorials

**Table 2 diagnostics-12-00919-t002:** Base-line characteristics of studies focusing on the detection of RNA biomarkers in oral biofluids. *m*, Male; *f*, Female; *H*, Periodontally Healthy; *CP*, Chronic Periodontitis; *P1*, Discovery Phase; *G*, Gingivitis; *P2*, Validation Phase; *exRNA*, Extracellular RNA; *miRNA*, MicroRNA; *GCF*, Gingival Crevicular Fluid.

Author	Year	Location	Study Design	*n* (*m/f*)	Mean Age (SD)	Classification	Biomarker	Biofluid
**Kaczor-Urbanowicz et al.**	2018	USA	PRoBE	*P1*— *H*: 50 (27/23) *G*: 50 (22/28) *P2*— *G*: 30 (13/17)	*P1*— *H*: 27.1 (5.67) *G*: 26.4 (6.77) *P2*— *G*: 28.2 (7.77)	*G*	*exRNA*	Saliva
**Micó-Martínez et al.**	2018	Spain	Cross-Sectional	*H*: 9 (4/5) *CP*: 9 (3/6)	*H*: 33.33 (12.05) *CP*: 50.44 (8.09)	*CP*	*miRNA*	*GCF*

**Table 3 diagnostics-12-00919-t003:** Base-line characteristics of studies analyzing protein-based biomarkers in oral biofluids.

Author	Year	Location	Study Design	*n* (*m/f*)	Mean Age (SD)	Classification	Biomarker	Biofluid
**Bostanci et al.**	2018	Turkey	Cross-sectional, Case–control	*P1*— *H*: 16 (5/11) *AP*: 17 (6/11) *CP*: 17 (7/10) *G*: 17 (9/8) *P2*— *H*: 20 (8/12) *AP*: 21 (5/16) *CP*: 21 (8/13) *G*: 20 (8/12)	*P1*— *H*: 34.13 (9.91) *AP*: 33.41 (5.40) *CP*: 44.88 (8.17) *G*: 33 (11.25) *P2*— *H*: 33.45 (6.35) *AP*: 33.67 (6.11) *CP*: 47.19 (7.01) *G*: 36.55 (5.16)	*AP*, *CP*, *G*	In total, 486 proteins were identified. MMP9, RAP1A, ARPC5, CLUS and DBMT1 showed greatest logical regression performance.	Saliva
**Chatzopoulos et al.**	2019	USA	Cross-sectional	*H*: 25 (6/19) *CP*: 25 (12/13)	*H*: 28.9 (10.2) *CP*: 57.9 (12.6)	*CP*	SOST, WNT-5a and TNF-α	*GCF*
**Cherian et al.**	2019	India	Cross-sectional	*H*: 30 *CP*: 30 *G*: 25	N/A (Range 18–45 years)	*CP*, *G*	Malondialdehyde	Salvia
**Hartenbach et al.**	2019	Brazil	Case–control	*H*: 10 (3/7) *CP*: 30 (16/14)	*H*: 29.9 (4.4) *CP*: 42.0 (2.6)	*CP*	In total, 473 proteins were identified, 223 were analyzed as they had *FDR* <5%.	Saliva

*m*, Male; *f*, Female; *P1*, Discovery Phase; *H*, Periodontally Healthy; *AP*, Aggressive Periodontitis; *CP*, Chronic Periodontitis; *G*, Gingivitis; *P2*, Validation Phase; *FDR*, False Discovery Rate; *GCF*, Gingival Crevicular Fluid.

**Table 4 diagnostics-12-00919-t004:** Base-line characteristics of studies analyzing metabolite-based biomarkers in oral biofluids.

Author	Year	Location	Study Design	*n* (*m/f*)	Mean Age (SD)	Classification	Biomarkers	Biofluid
**Chen et al.**	2018	China	Cross-sectional	*H*: 20 (10/10) *GAgP*: 20 (9/11)	*H*: 25.7 (4.5) *GAgP*: 28.4 (4.3)	*GAgP*	In total, 349 metabolites were detected in *GCF*.	*GCF*
**Pei et al.**	2020	China	Cross-sectional	*H*: 28 (9/19) *GCP*: 30 (13/17)	*H*: 35.7 (N/A) *GCP*: 39.0 (N/A)	*GCP*	In total, 147 metabolites were identified from samples obtained from both cases and controls.	*GCF*
**Romano et al.**	2018	Italy	Cross-sectional	*H*: 39 (25/14) *GCP*: 33 (21/12) *GAgP*: 28 (18/10)	*H*: 46.6 (8.2) *GCP*: 50.5 (8.9) *GAgP*: 31.1 (4.6)	*GCP, GAgP*	Twenty-two metabolites were identified and analyzed.	Saliva
**Rzeznik et al.**	2017	France	Cross-sectional	*H*: 25 (9/16) *P*: 26 (10/16) *CP*: 18 *GAgP*: 8	*H*: 40.7 (12.4) *P*: 42.4 (12.8)	*P (CP, GAgP)*	Eleven metabolites were identified as being discriminatory between health and disease.	Saliva

*m*, Male; *f*, Female; *H*, Healthy; *GAgP*, Generalized Aggressive Periodontitis; *GCP*, Generalized Chronic Periodontitis; *P*, Periodontitis; *CP*, Chronic Periodontitis; *GCF*, Gingival Crevicular Fluid.

**Table 5 diagnostics-12-00919-t005:** Base-line characteristics of studies analyzing inflammatory biomarkers in oral biofluids.

Author	Year	Location	Study Design	*n* (*m/f*)	Mean Age (SD)	Classification	Biomarker	Biofluid
**Hong et al.**	2020	South Korea	Cross-sectional	*H*: 15 (8/7) *G*: 85 (38/47)	*H*: 34.93 (15.79) *G*: 32.65 (12.21)	*G*	MMP-8, MMP-9, lactoferrin, cystatin C, MPO, platelet-activating factor, cathepsin B, pyridinoline cross-linked carboxterminal telopeptide of type 1 collagen	Saliva and *GCF*
**Inönü et al.**	2020	Turkey	Cross-sectional	*H*: 45 (16/29) *G*: 50 (16/29) *CP*: 5 (26/24) *GAgP*: 40 (15/25)	*H*: 28.0 *G*: 24.5 *CP*: 43.5 *GAgP*: 28.0 *	*G*, *CP*, *GAgP*	Del-1, IL-17, LFA-1	Saliva
**Nalmpantis et al.**	2020	Greece	Cross-sectional	*H*: 48 (18/30) *CP*: 53 (33/20)	*H*: 50.8 (9) *CP*: 52.0 (8)	*CP*	Azurocidin	*GCF*
**Sai Karthikeyan et al.**	2020	India	Cross-sectional	*H*: 10 *G*: 10 *P*: 10 (17/13)	*H*: 22.2 (3.46) *G*: 35.7 (8.12) *P*: 42.4 (6.84)	*G*, *P*	Soluble CD163	*GCF*
**Taşdemir et al.**	2020	Turkey	Cross-sectional	*H*: 20 (12/8) *G*: 20 (11/9) *CP*: 20 (10/10)	*H*: 29.40 (6.63) *G*: 29.30 (11.53) *CP*: 47.75 (9.88)	*G*, *CP*	suPAR, Galectin-1, TNF alpha	Saliva and *GCF*

*m*, Male; *f*, Female; *H*, Healthy; *G*, Gingivitis; *CP*, Chronic Periodontitis; *GAgP*, Generalized Aggressive Periodontitis; *P*, Periodontitis; *GCF*, Gingival Crevicular Fluid. * Age as a median figure, not mean measurement. SD not provided.

**Table 6 diagnostics-12-00919-t006:** Disease classification, number of examiners, sample collection method, sample analysis technique and brief outcome of studies analyzing RNA biomarkers.

Author	Year	Classification	No. of Examiners	Sample Collection	Analysis	Outcome
**Kaczor-Urbanowicz et al.**	2018	*P1*— *H*: *MBI* < 5% and *PPD* < 4 mm *G*: *MBI* > 5% and *PPD* < 4 mm *P2*— ≥20 Natural Crowned teeth, *GI* ≥ 1.0 and *PI* ≥ 1.5	2. Change in examiner at week 6 which may affect clinical parameters	Unstimulated whole saliva samples were collected prior to clinical examination. Hygiene measures, eating and drinking were avoided 1 h before collection. All subjects rinsed with 10 mL tap water 10 min before sample collection. Approx. 5 mL saliva was collected in a 5–10 min period. The samples were stored in sterile tubes and held on ice until processing approx. 1 h after collection.	*P1*—Rneasy Micro Kit *P2*—*RT*-*qPCR*	Increase of four exRNAs and decrease of four *exRNAs* in gingivitis (*exRNAs* identified following the discovery phase), with four potentially discriminatory of health.
**Micó-Martínez et al.**	2018	*H*: *PPD* < 3 mm, *CAL* < 3 mm and no radiographic evidence of bone deterioration. *CP*: At least 1 single rooted tooth with *PPD* ≥ 5 mm and *CAL* ≥ 6 mm. Classification based on the guidelines from The Classification of Periodontal Diseases and Conditions Armitage 1999.	1	Prior to collection, supragingival plaque was removed and cotton balls along with aspiration were used to prevent salvia contamination. *GCF* was sampled from a single-rooted tooth. PerioPaper^®^ was placed in the gingival sulcus until resistance was noted and left for 30 s. Contaminated samples were discarded, and the procedure was repeated. Samples were stored in EP tubes at −80 °C.	miRNeasy Serum/Plasma Kit	miR-1226 identified as having potential diagnostic capabilities.

*P1*, Discovery Phase; *P2*, Validation Phase; *H*, Healthy; *G*, Gingivitis; *MBI*, Marginal Bleeding Index; *PPD*, Pocket Probing Depth; *GI*, Gingival Index; *PI*, Plaque Index; *CAL*, Clinical Attachment Loss; *CP*, Chronic Periodontitis; *GCF*, Gingival Crevicular Fluid; *EP*, Eppendorf; *RT-qPCR*, Real Time Quantitative Reverse Transcription Polymerase Chain Reaction.

**Table 7 diagnostics-12-00919-t007:** Disease classification, number of examiners, sample collection method, sample analysis technique and brief outcome of studies analyzing protein biomarkers.

Author	Year	Classification	No. of Examiners	Sample Collection	Analysis	Outcome
**Bostanci et al.**	2018	*H*: *PPD* > 3 mm, *CAL* > 2 mm, mean *BOP* < 15% and no detectable bone loss. AP: >16 teeth, *PPD* > 6 mm, *CAL* > 5 mm on 8 or more teeth and bone loss of >30% of root length affecting more than 3 teeth. *CP*: *PPD* > 6 mm, *CAL* > 5 mm, *BOP* > 63% and bone loss of >50% in at least 2 quadrants. *G*: Varying inflammation, *CAL* < 2 mm, mean *BOP* > 50% and no bone loss. Classification based on The Classification of Periodontal Diseases and Conditions Armitage 1999.	1	Whole saliva samples collected prior to clinical classification between 08:00 and 10:00. Participants asked not to undertake hygiene measures, eat or drink for 2 h prior to sample collection. The participants rinsed with water for 2 min, waited for 10 min then expectorated for 5 min into a sterile tube. The samples were held on ice until analysis.	*P1*—*LC-MS P2*—*QTRAP* 5500 and Nano-*LC-2D HPLC*	*P1* of this study identified almost 200 proteins with diagnostic potential. *P2* yielded a list of five proteins with both high sensitivity and specificity for periodontal disease diagnosis.
**Chatzopoulos et al.**	2019	*G1*— *H*: *PPD* ≤ 4 mm, *CAL* ≤ 2 mm and no radiographic evidence of bone loss. *CP*: *PPD* ≥ 5 mm, *CAL* ≥ 4 mm and radiographic imagining presenting alveolar bone loss ≥40%. *G2*— *H*: *PPD* 1–3 mm, *CAL* ≤ 1 mm and *BOP* ≤ 15%. *CP*: *PPD* ≥ 4 mm, *CAL* ≥ 3 mm and presence of *BOP*.	2	Prior to *GCF* collection, supragingival plaque removed and a gentle stream of air applied for approx. 3–5 s to the interproximal surface. PerioPaper^®^ was placed in the crevice until resistance was felt and left for 30 s. Contaminated samples were discarded, and the procedure was repeated. The papers were stored at −80 °C until processing.	*ELISA*	SOST and WNT-5a identified as having good diagnostic capabilities in generalized, moderate and severe periodontitis, but not localized periodontitis.
**Cherian et al.**	2019	*H*: No history of periodontal disease. *CP*: At least 4 teeth exhibiting *PPD* ≥ 4 mm, *CAL* ≥ 4 mm and *BOP* evident. *G*: *BOP* evident.	N/A	Prior to clinical examination, whole saliva samples collected between 09:00 and 12:00. Samples collected approximately 2 h after food. Approx. 2 mL of salvia collected into disposable tubes and centrifuged immediately. Sample analysis was completed immediately after collection.	Spectrophotometry	This study concluded that malondialdehyde levels are significantly different between *H* and *CP* (*p* < 0.001), but not between *H* and *G* (*p*~0.089).
**Hartenbach et al.**	2019	*H*: *PPD* 1.9 mm, *CAL* 2.0 mm, *BOP* 4.4%, *GI* 4.9%, *PL* 20.9% and *SC* 13.2%. *CP*: *PPD* 2.5 mm, *CAL* 2.7 mm, *BOP* 28.6%, *GI* 14.1%, *PL* 38.4% and *SC* 28.7%. ¹	1	Saliva samples obtained in the morning period at least 2 h prior to dental hygiene measures. Participants were asked to rest for 15 min and saliva was stimulated using Parafilm M^®^. Approximately 1 mL of saliva was obtained from each participant in a sterile plastic tube. The samples were pooled in pairs of individuals with similar age. They were held on ice until centrifugation and then frozen at −80 °C until analyzed.	*LC-MS*	Few specific biomarkers were increased in *CP* compared to *H*. Therefore, diagnosis may be based on decreased levels of several proteins.

*H*, Health; *PPD*, Pocket Probing Depth; *CAL*, Clinical Attachment Level; *BOP*, Bleeding on Probing; *AP*, Aggressive Periodontitis; *CP*, Chronic Periodontitis; *G*, Gingivitis; *P1*, Discovery Phase; *P2*, Validation Phase; *GCF*, Gingival Crevicular Fluid; *LC-MS*, Liquid Chromatography Mass Spectroscopy; *HPLC*, High-Performance Liquid Chromatography; *G1*, Normal Classification; *G2*; Strict Classification; *ELISA*, Enzyme-Linked Immunosorbent Assay; *SC*, Supragingival Calculus. ¹ Mean levels of parametric clinical measurements only recorded.

**Table 8 diagnostics-12-00919-t008:** Disease classification, number of examiners, sample collection method, sample analysis technique and brief outcome of studies analyzing metabolite-based biomarkers.

Author	Year	Classification	No. of Examiners	Sample Collection	Analysis	Outcome
**Chen et al.**	2018	*PPD*, *CAL* and radiographic imaging were measured. *H*: *PPD* ≤ 3 mm *CAL* = 0 mm. *GAgP*: *CAL* ≥ 5 mm. Classification based on The Classification of Periodontal Diseases and Conditions Armitage 1999.	1	Prior to *GCF* sample collection, supragingival plaque was removed and the tooth was air-dried. PerioPaper^®^ was inserted into the gingival sulcus for 30 s and stored in EP tubes at −80 °C. Contaminated strips were discarded.	Gas Chromatography Mass Spectrometry	Noradrenaline, uridine, dehydroascorbic acid, ribose and methionine levels were all elevated in those with *GAgP*. Levels of 2-ketobutyric acid, glycine-d5, thymidine and ribose-5-phosphate were all reduced in those with *GAgP*.
**Pei et al.**	2020	*H*: *PPD* ≤ 3 mm, *CAL* < 1 mm. *GCP*: *PPD* ≥ 4 mm, *CAL* ≥ 3 mm, *BOP* present.	1	Before sampling, supragingival plaque was removed and the tooth was air-dried. Samples were collected using PerioPaper^®^ which was gently inserted into the gingival sulcus and left for 30 s. The papers were stored in EP tubes and stored at −80 °C.	Gas Chromatography Mass Spectrometry	In total, 17 metabolites were analyzed. Glycine-d5, *N*-carbamylglutamate 2 and fructose were increased in *GCP* compared to *H*. Lactamide, *O*-phosphoserine and 1-monopalmitin were decreased in *GCP* compared to *H*. Pyrimide, d-glutamine and d-glutamate metabolism were increased in *GCP* compared to *H*. Combination of citramalic acid and *N*-carbamylglutamte produced the most accurate diagnostic measure of disease.
**Romano et al.**	2018	*H*: *PPD* ≤ 3 mm, *CAL* ≤ 3 mm, no radiographic evidence of bone loss and <15% of *BOP*. *GCP*: *PPD* ≥ 30% sites, *CAL* > 5 mm and presence of BOP. *GAgP*: At least one site with *PPD*, *CAL* > 5 mm and at least one of *PPD* or *CAL* >5 mm. Classification based on the World Workshop in Periodontology 1999.	2	Unstimulated saliva samples were collected at least 24 h after clinical examination between 08:00 and 10:00. Participants were asked not to undertake hygiene measures 1 h prior to collection. Participants was asked not to force salvation and approx. 1 ml was collected in a sterile tube over 10 min and frozen immediately.	NMR Spectroscopy	Several metabolites were identified as being significantly different between *H* and disease. Lower levels of pyruvate, lactate and *N*-acetyl groups in *GCP* and lower levels of pyruvate, lactate, *N*-acetyl groups and sarcosine in *GAgP* versus H. Higher levels of phenylalanine and tyrosine in both *GCP* and *GAgP* compared to *H*. Phenylalanine metabolism and pyruvate metabolism pathways were identified as being most significant in discrimination between *H* and disease.
**Rzeznik et al.**	2017	*P*: ≥ 2 sites with *PPD* ≥ 4 mm (not on the same tooth), *CAL* ≥ 3 mm or one site with *PPD* ≥ 5 mm. *PPD* 3.8 mm, *CAL* 4.1 mm, *BOP* 35.0%, *PCR* 61.2%, Affected sites 48.5%, *DMF* 8.23. ¹ Classification according to The Classification of Periodontal Diseases and Conditions Armitage 1999.	1	Saliva was collected prior to clinical examination between 09:00 and 11:00 and stimulated using paraffin wax. Participants were asked not to eat, drink, chew gum or brush teeth for 2 h prior to collection. Approx. 10 mL of saliva was collected for 5 min. The pH was recorded immediately, and the samples were stored at −25 °C until analysis.	*HNMR* Spectroscopy	*HNMR* analysis identified increases in Butyric acid in both *CP* and *GAgP*, while levels of both lactic acid and GABA were deceased in *CP* and *GAgP* compared with H. No metabolite discriminated against *CP* and *GAgP.* Combination of three aforementioned metabolites displayed good diagnostic capabilities.

*H*, Health; *PPD*, Pocket Probing Depth; *CAL*, Clinical Attachment Loss; *GAgP*, Generalized Aggressive Periodontitis; *GCF*, Gingival Crevicular Fluid; *EP*, Eppendorf; *BOP*, Bleeding on Probing; *GCP*, Generalized Chronic Periodontitis; *GAgP*, Generalized Aggressive Periodontitis; *AUC*, Area Under the Curve; *P*, Periodontitis; *PCR*, Plaque Control Record; *DMF*, Decay Missing Filled; *HNMR*, Proton Nuclear Magnetic Resonance. ¹ Mean levels of parametric clinical measurements only recorded.

**Table 9 diagnostics-12-00919-t009:** Disease classification, number of examiners, sample collection method, sample analysis technique and brief outcome of studies analyzing inflammatory biomarkers.

Author	Year	Classification	No. of Examiners	Sample Collection	Analysis	Outcome
**Hong et al.**	2020	*PPD*, *CAL*, *BOP*, *PI* and *GI* were measured. *H*: *CAL* 2.55 mm *BOP* 5.56% *PI* 0.13 *GI* 0.39 *G*: *CAL* 2.60 mm *BOP* 26.96% *PI* 0.53 *GI* 0.96. ¹ Those with >10% *BOP* were diagnosed with *G*. Classification according to the World Workshop on the Classification of Periodontal and Peri-Implant Diseases and Conditions 2017.	1	Participants were asked to fast for 8 h before *GCF* collection. Site of collection was air-dried and supragingival plaque was removed prior to collection. Paper points were inserted into the crevice and left for 30 s. Contaminated samples were discarded. Samples were stored at 4 °C overnight and centrifugation occurred for 5 min at 4 °C. Samples were then stored at −80 °C until analysis. Patients rinsed with pure water and whole saliva samples were obtained by holding a cotton roll in the mouth for 60 s. Samples were centrifuged immediately for 10 min and then stored at −80 °C until analysis.	*ELISA*	MMP-8 and MPO levels displayed significant differences between *H* and *G*, therefore suitable for disease diagnosis. Further research required to develop and chairside diagnostic test.
**Inönü et al.**	2020	*PPD*, *CAL*, *BOP*, *PI*, *GI* and *BMI* were measured. *H*: *PPD* ≤ 3 mm *BOP* <10% of sites, no bone loss. *G*: *PPD* ≤ 3 mm, *GI* > 0, no bone loss. *CP*: *PPD* ≥ 5 mm *CAL* ≥ 4 mm, ≥50% bone loss. *GAgP*: Severe interproximal attachment loss impacting ≥3 permanent teeth and presented with symptoms <30 years old. Classification based The Classification of Periodontal Diseases and Conditions Armitage 1999.	N/A	Unstimulated whole saliva samples were collected prior to clinical examination. Participants were asked not to consume any food for 1 h before sampling. Samples were collected into a plastic tube during a 5 min period. The sample was aspirated from a 5 mL syringe and 3 mL was collected and stored at −80 °C until analysis.	*ELISA*	Del-1 levels were increased in both *H* and *G* compared to *CP* and *GAP*. IL-17 levels were lower in both *H* and *G* in comparison to *CP* and *GAP*; however, IL-17 levels were more elevated in *G* than *H*. LFA-1 levels were elevated in *G*, *CP* and *GAP* compared to *H*. Further studies required to determine the efficacy of these biomarkers in disease diagnosis.
**Nalmpantis et al.**	2020	*PPD*, *CAL*, *BOP*, *PL* and *REC* were measured. *H*: *PPD* < 3 mm *BOP* <10% *CP*: ≥30% of teeth with *CAL* ≥ 5 mm. Classification based The Classification of Periodontal Diseases and Conditions Armitage 1999.	3	Cotton rolls were used to prevent saliva contamination. The site was air-dried and supragingival plaque was removed. PerioPaper^®^ was inserted into the crevice at least 1–2 mm and left for 30 s. *GCF* samples were pooled (4 samples per participant). Samples were immediately frozen using liquid nitrogen at −196 °C and stored at –80 °C until analysis.	*ELISA*	Levels of azurocidin were significantly elevated in those with *CP* compared to *H* (AUC = 0.915). Further research required in order to determine the value of azurocidin as a biomarker for disease.
**Sai Karthikeyan et al.**	2020	*PPD*, *CAL*, *BOP* and *GI* were measured. *H*: *BOP* (-), *CAL* (-) *G*: *BOP* (+) Inflammation, however *CAL* (-) *P*: *CAL* ≥ 3 mm bone loss evident	N/A	*GCF* samples were collected 1 day after clinical examination. Site was air-dried and cotton rolls were used to prevent salvia contamination. Supragingival plaque was removed. A 10 µL micropipette was inserted into the crevice and at least 5 µL of *GCF* was collected. Contaminated samples were discarded. The samples were stored in air-protected plastic vials and at −70 °C until analysis.	*ELISA*	Levels of CD163 were significantly increased in disease, both *G* and *P* compared to *H*. Further research into the development of a chairside test for CD163 is required in order to diagnose disease.
**Taşdemir et al.**	2020	*PPD*, *CAL*, *BOP*, *PI* and *GI* were measured. *H*: *PPD* ≤ 3 mm, *CAL* = 0 < 20% *BOP*, *GI* < 1 and no bone loss. *CP*: *PPD* ≥ 5 mm *CAL* ≥ 4 mm, *BOP* (+), GI = 2 and bone loss of >40%. *G*: *PPD* ≤ 3 mm *CAL* = 0 *BOP* ≥ 20%, *GI* = 2 and no bone loss. Classification based on The Classification of Periodontal Diseases and Conditions Armitage 1999.	1	*GCF* samples were collected 1–2 days after clinical diagnosis. Dental aspirator and cotton rolls were used to avoid salvia contamination. Supragingival plaque was removed. PerioPaper^®^ was inserted into the crevice and left for 30 s. Contaminated samples were discarded. Samples were pooled (4 samples per participant) and stored at −80 °C until analysis. Saliva samples were collected before 12:00. Participants were asked not to eat or drink for 1 h before collection. Unstimulated whole saliva was collected by expectorating into a plastic tube. Samples were transferred to a sterile syringe and centrifugated immediately for 10 min at room temp. In total, 0.5 mL of the sample was then added to PP tubes and stored at −80 °C until analysis.	*ELISA*	Increased *GCF* levels of suPAR and galectin-1 were identified between disease and health. Salvia levels of suPAR were higher in *CP* compared to *G* and *H*. This study concluded, suPAR may be a useful biomarker in disease diagnosis.

*H*, Health; *PPD*, Pocket Probing Depth; *CAL*, Clinical Attachment Loss; *BOP*, Bleeding on Probing; *PI*, Plaque Index; *GI*, Gingival Index; *G*, Gingivitis; *GCF*, Gingival Crevicular Fluid *ELISA*, Enzyme-Linked Immunosorbent Assay; *BMI*, Body Mass Index; *CP*, Chronic Periodontitis; *GAgP*, Generalized Aggressive Periodontitis; *REC*, Gingival Recession; *AUC*, Area Under the Curve; *PP*, Polypropylene. ¹ Mean levels of parametric clinical measurements only recorded.

## Data Availability

The authors confirm that the data supporting the findings of this study are available within the article and/or its Appendix A.

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
