# Peer review of "The Role of Epigenetic and Biological Biomarkers in the Diagnosis of Periodontal Disease: A Systematic Review Approach"

_diagnostics, 2022, doi:10.3390/diagnostics12040919_

Round 1

Reviewer 1 Report

Dear authors,

The paper written by Faulkner et al. Approaches approaches a very interesting topic regarding the epigenic and biological biomarkers related to periodontal diagnosis.

I hope that my remarks will help the authors to improve the quality of the paper.

Line 27 – please add more keywords;

Lines 76-78 – please add citation and elaborate about electronic probing;

Lines 58-60 – requires citation;

Lines 84-85 – please elaborate regarding the roles of salivary MMPs and PGE2in quantifying periodontal inflammation;

Table 1 – Why did you exclude peri-implant disease?

Line 221- statistical method should have been described here;

Tables 2 and 6 – For me it’s a bit is a bit doubtful how the saliva was preserved in the study conducted by Kaczor-Urbanowicz. Processing all samples in one hour from collection is in my opinion a bit unrealistic. Usually samples are collected, frozen and processed when a significant number is gathered for more efficiency.

Discussion part: please insert a section about MMPs and a potential relationship to mRNA as biomarkers.

Line 710 - Please rephrase the Conclusion section in order to underline the practical, clinical impact and to emphasise future perspectives.

Please receive my kindest regards!  

Author Response

Dear Reviewer,
Thank you for the opportunity to submit a revised draft of my manuscript titled The Role of Epigenetic and Biological Biomarkers in the Diagnosis of Periodontal Disease: A systematic review approach (Manuscript ID: diagnostics-1626437) to Diagnostics. We appreciate the time, valuable comments, and comprehensive review of our manuscript. We are grateful for the insightful comments and suggestions on our paper. We have been able to incorporate changes and addressed the concerns in the revised manuscript. All changes made can tracked in the attached revised manuscript. In this letter, the changes are highlighted in blue. Please find a point-by-point response to the comments in the attachment. 

Thank you.

Reviewer 2 Report

The present review paper is well written and designed. I suggest to the authors to add a graphical abstract with some figures to attract more views.

Author Response

Dear Reviewer,

We appreciate your valuable time, comments and comprehensive review of our paper.

Comment 1: The present review paper is well written and designed. I suggest to the authors to add a graphical abstract with some figures to attract more views.

Thank you for the suggestion, we agree a graphical abstract will attract more views. However, we currently are unable to settle on one figure that will give a comprehensive summary of our work. We anticipate that the current abstract provided in lines 12 to 26 which is less than 200 words will be a short read for viewers.

In addition to the above comments, all references have been updated in the text and in the references section (lines 775 to 794).

Round 2

Reviewer 1 Report

Dear authors,

Thank you very much for the revised version and I think that it respected most of my suggestions.

Best regards!

Author Response

Thank you for your valuable time in reviewing our manuscript.